Production and characteristics of fish protein hydrolysate from parrotfish (Chlorurus sordidus) head

http://orcid.org/0000-0003-4180-1325 Prihanto Asep A. 1 2 asep_awa@ub.ac.id
http://orcid.org/0000-0002-5368-419X Nurdiani Rahmi 1 2
Bagus Annas D. 2
1 Department of Fishery Product Technology, Faculty of Fisheries and Marine Science, Brawijaya University , Malang, East Java , Indonesia
2 BIO-SEAFOOD Research Unit, Brawijaya University , Malang, East Java , Indonesia
Sotelo-Mundo Rogerio
Electronic publication date: 2019 Dec 20
Publication date: 2019
Volume: 7
Electronic Location ID: e8297
Received 2019 Oct 4; Accepted 2019 Nov 26
Copyright: © 2019 Prihanto et al.
Copyright year: 2019
Copyright holder: Prihanto et al.
License: This is an open access article distributed under the terms of the Creative Commons Attribution License, which permits unrestricted use, distribution, reproduction and adaptation in any medium and for any purpose provided that it is properly attributed. For attribution, the original author(s), title, publication source (PeerJ) and either DOI or URL of the article must be cited.
License URL: https://creativecommons.org/licenses/by/4.0/

Keywords: Endogenous enzyme, Parrotfish, Fish-by product, Fish protein hydrolysate, Antioxidant

Funding: Ministry of Research, Technology and Higher Education, the Republic of Indonesia 167/SP2H/LT/DRPM/2019 Brawijaya University for Basic Research on Higher Education 330.32/UN10.C10/PN/2019 This work was supported by the Ministry of Research, Technology and Higher Education, the Republic of Indonesia (No. 167/SP2H/LT/DRPM/2019), and the Brawijaya University for Basic Research on Higher Education (No. 330.32/UN10.C10/PN/2019). The funders had no role in study design, data collection and analysis, decision to publish, or preparation of the manuscript.

==============================
Background

Fish byproducts are commonly recognized as low-value resources. In order to increase the value, fish byproducts need to be converted into new products with high functionality such as fish protein hydrolysate (FPH). In this study, FPH manufactured from parrotfish (Chlorurus sordidus) heads using different pH, time and sample ratio was investigated.

Methods

Hydrolysis reactions were conducted under different pHs (5, 7, and 9) and over different durations (12 and 24 h). Control treatment (without pH adjustment (pH 6.4)) and 0 h hydrolsisis duration were applied. Hydrolysates were characterized with respect to proximate composition, amino acid profile, and molecular weight distribution. The antioxidant activity of the hydrolysate was also observed.

Results

The pH and duration of hydrolysis significantly affected (p < 0.05) the characteristics of FPH. The highest yield of hydrolysate (49.04 ± 0.90%), with a degree of hydrolysis of 30.65 ± 1.82%, was obtained at pH 9 after 24 h incubation. In addition, the FPH had high antioxidant activity (58.20 ± 0.55%), with a high level of essential amino acids. Results suggested that FPH produced using endogenous enzymes represents a promising additive for food and industrial applications.

Introduction

Parrotfish (Chlorurus sordidus) are one of the most important fish commodities in Indonesia (Adrim, 2010). Parrotfish have unique and exceptional arrangements of teeth and body shape (Chen, 2002). In 2014, parrotfish fishing production increased by 18.8% (76 tons) compared to all reef fish fisheries in the Asian region (Food & Agriculture Organization (FAO), 2015). In Indonesia, a total of 1.8 ton parrotfish production was recorded for 2019 (Ministry of Marine Affair & Fisheries, 2019). An increase in the number of catch means an increase in the amount of byproduct processing, as processing requires the removal of bones, skin, head, scales, and viscera. Out of all the other body parts, the head accounts for approximately 19% of the total fish processing-byproducts from fillet processing (Anil, 2017). Several parts of by product such as scales and bone was applied as gelatin (Herpandi, Huda & Adzitey, 2011). In contrast, fish head was still underutilized.

Fish byproducts, commonly recognized as low-value resources, can be further developed into products with high economic value if handled and processed appropriately (Hapsari & Welasi, 2013). In general, fish byproducts contain many elements, such as nitrogen, phosphorus, potassium, and others, which are the constituents of proteins and fats (Lepongbulan, Tiwow & Diah 2017). Thus, the protein fraction of byproducts can be utilized for the production of fish protein hydrolysate (FPH) with desirable functionality. In addition, FPH has reported to exhibit bioactive properties, such as antihypertensive, antioxidant (Yang et al., 2011), antithrombotic (Qiao et al., 2018), anticancer, and immunomodulatory activities (Kim & Mendis, 2006).

Fish protein hydrolysate can be manufactured from the decomposition of fish proteins into simple peptides (2–20 amino acids) through hydrolysis by adding enzymes, acids, or bases (Nurilmala, Nurhayati & Roskananda, 2018). The characteristics and quality of FPH are highly influenced by several factors, including the type of proteases or chemicals used, temperature, pH, and duration of hydrolysis (Nazeer & Kulandai, 2012).

In previous studies, FPH was developed using various fisheries byproducts, including cod head waste (Himonides, Taylor & Morris, 2011), catfish (Nurilmala, Nurhayati & Roskananda, 2018), tuna (Bougatef et al., 2012; Herpandi, Huda & Adzitey, 2011), Sardinella (Jeevitha, Priya & Khora, 2014), and tilapia (Srikanya et al., 2017). Nevertheless, the production of FPH from parrotfish byproducts remains limited. This study aimed to determine the characteristics of protein hydrolysates from parrotfish (C. sordidus) heads, extracted at different pHs and hydrolysis duration periods.

Materials and Methods

Materials

All materials used in this experiment were of analytical grade and were purchased from Merck (Darmstadt, Germany, USA). Parrotfish (C. sordidus) heads with the average weight of 250 ± 18 gr were obtained from a local fish processing plant (PT. Alam, Surabaya, Indonesia). The heads were transported to the laboratory using a storage box maintained at 4 °C.

Preparation of fish protein hydrolysate

Preliminary experiments on the optimum water: substrate ratios were conducted to obtain the highest yield and antioxidant activity of hydrolysate. Fish heads were crushed in Philips-Food Processor, model HR7627, 650 W, capacity 2.1 L. Briefly, 20 g of minced fish head was mixed with dH2O in ratios of 1:0, 1:1, 1:2, and 1:3 (w/v). Hydrolysis for 18 h was conducted using an orbital shaker at 150 rpm at temperature of 30 ± 2 °C. Next, the mixture was centrifuged at 3,000 rpm for 30 min. Each layer formed after centrifugation was separated and weighed. The liquid protein layer was also analyzed for antioxidant activity. The data were obtained by triplicate analysis.

The effect of pH and duration of hydrolysis on the characteristics of FPH was investigated as per a modified method of that previously described (Sabtecha, Jayapriya & Tamilselvi, 2014; Nurdiani et al., 2016). Minced parrotfish head (20 g) was mixed with dH2O (1:2 w/v). The pH of the mixture was adjusted to 5, 7, and 9, and the hydrolysis was conducted for 12 and 24 h. Samples without pH adjustment served as the control (pH 6.4). A similar procedure as the preliminary experiments was carried out to obtain hydrolysate.

Process optimization

An optimum process was obtained by analyzing the data using response surface methodology (RSM). An overlaid contour plot was applied to select the best hydrolysis conditions for FPH. Minitab version 18 was used for all statistical analysis.

Yield

The yield of protein hydrolysate products is defined as the percentage of the number of hydrolysate products produced against the raw materials used before hydrolysis. Yield is calculated as per the following formula: Yield=AB×100%

where A = final weight of hydrolysate (after centrifugation) (g), and B = initial weight of the sample after mixing (before incubation) (g).

Antioxidant assay (DPPH radical scavenging activity)

The antioxidant activity of FPH was examined according to a modified protocol described by Donkor et al. (2012). As much as 100 μL of liquid protein was added to 3,900 μL 0.075 mM 2, 2-Diphenyl-1-picrylhydrazyl (DPPH) in 95% methanol; the mixture was kept in the dark for 1 h. The absorbance value of the solution was measured at a wavelength of 517 nm using an ultraviolet-visible spectrophotometer. Antioxidant activity was calculated using the following equation: %antioxidantactivity=[blankabsorbance−sampleabsorbanceblankabsorbance]×100%

Proximate analysis

Protein, fat, water content, and ash analyses were performed according to the method described by AOAC (2005). Protein was analyzed following the Kjeldahl method, and fat was analyzed using the Soxhlet method. Ash was determined by heating the samples in a furnace at 550 °C for 8–12 h.

Degree of hydrolysis

A slightly modified method of that described by Hoyle & Merritt (1994) was employed for the DH analysis. Liquid FPH (two mL) was combined with Trichloroacetic acid 20% (v/v); the aliquot was left for 30 min prior to centrifugation (5,000 rpm, 30 min). The supernatant was decanted and analyzed for nitrogen content following the Kjeldahl method (AOAC, 2005). DH was calculated using the following formula: DegreeofHydrolysis(DH)=TCA−solublenitrogenTotalnitrogeninsample×100%

Molecular weight analysis (SDS–PAGE)

Fish protein hydrolysate molecular weight was determined by sodium dodecyl sulfate-polyacrylamide gel electrophoresis (SDS–PAGE), based on the Laemmli method (Laemmli, 1970). SDS-PAGE analysis utilized a 12% separating gel and 4% stacking gel. Mixed samples and loading buffers, as much as 30 µL, were run at 20 mA and 100 V for 3 h. The gel was then stained with staining solution Coomassie Brilliant Blue (CBB) R-250 1 g, methanol 450 mL, glacial acetic acid 100 mL, and distilled water 450 mL. The stained gel was subsequently de-stained using the same solution without CBB R-250.

Free amino acid analysis

Fish protein hydrolysate free amino acid profiles were determined according to a slightly modified method of that described by Boogers et al. (2008). Ultra-High Performance Liquid Chromatography (UPLC), using an Acquity system (Waters), was utilized for free amino acid analysis. Sample (0.50 mL) was pipetted into a 100 mL volumetric flask, and 2.0 mL of alpha amino butyric acid 10 mM internal standard solution was added. The solution was diluted to the limit mark with 0.1 N HCl, before being homogenized. Next, the solution was filtered through a 0.22 μm membrane filter. Ten microliters of the solution was added to 70 μL of AccQ-Fluor Borate. After that, up to 20 μL fluorine reagent A was added, before being vortexed, and allowed to stand for 1 min. One microliter of sample solution was injected into the UPLC system (ACCQ-Tag Ultra C18, fluid rate system of 0.7 mL per minute, the column temperature was maintained at 55 °C, and a photodiode array detector at a wavelength of 260 nm.

Statistical analysis

All data and RSM optimizations were analyzed by using Minitab 18 Statistical software (Minitab Pty Ltd., Sydney, NSW, Australia). Except data for optimization, all data obtained were subjected to one-way analysis of variance, followed by post-hoc test (Tukey analysis). Data are presented as the mean from three independent experiment ± SD of the results.

Results

Proximate composition of parrotfish heads

The proximate composition of minced parrotfish (C. sordidus) heads is listed in Table 1. The protein content, at 20.37 ± 2.33%, was higher than salmon and Mackarel head. The fat content (3.92%) was slightly higher than Mackarel fish 3.70%, and far lower than salmon (17.4%). The water content (71.68 ± 2.33%) was higher than that of salmon (65.9%) but lower Mackarel fish (65.9%).

Table 1 Proximate composition of minced Parrotfish, Salmon, and Nile.

Parameter	Parrotfish*	Salmon**	Nile***	
Carbohydrate (%)	0.52 ± 0.13	–	37.78	
Protein (%)	20.37 ± 2.33	11.90	29.80	
Fat (%)	3.92 ± 0.38	17.40	3.10	
Water (%)	71.68 ± 1.87	65.90	5.70	
Ash (%)	4.19 ± 0.66	4.30	21.80	
Notes:

* This study.

** Wu et al. (2011).

*** Kefas et al. (2014).

Fish protein hydrolysate from parrotfish heads

Five layers were formed after centrifugation. The first layer was oil/fat, followed by light lipoprotein, soluble protein, fine particles, and coarse particles layers (Fig. 1A). Soluble protein layers were carefully separated and collected (Fig. 1B). The yield and antioxidant activity of liquid/soluble protein were measured. The soluble protein layer was also spray-dried (Fig. 1C).

Figure 1 FPH from parrotfish head.

(A) Formed layers after centrifugation. (B) Collected soluble protein layer. (C) Dried FPH.

Effect of substrate: water ratio on the yield and antioxidant activity of soluble protein

The ratio of minced head: dH2O significantly affected (p < 0.05) the yield and antioxidant activity of the FPH produced, as seen in Fig. 2. Among the four ratios (1:0, 1:1, 1:2, and 1:3), the highest yield and antioxidant activity were obtained from the ratio of 1:2 (w/v), with values of 42.70 ± 0.70 and 51.50 ± 0.90%, respectively. The ratio of 1:0 generated the lowest yield and antioxidant activity.

Figure 2 Yield and antioxidant activity of FPH.

Effect of pH and hydrolysis duration on FPH characteristics

The characteristics of FPH from parrotfish head hydrolyzed at various pH and time durations are shown in Table 2.

Table 2 Characteristics of FPH from parrotfish heads with different pH and hydrolysis duration.

Parameter	Control*	5	7	9	
0**	12	24	0**	12	24	0**	12	24	0**	12	24	
Yield	5.78 ± 0.85a	39.15 ± 0.87b	47.48 ± 1.29cd	5.50 ± 2.03a	37.73 ± 0.92b	45.4 ± 1.17c	4.96 ± 0.72a	36.36 ± 1.03b	48.37 ± 0.63cd	6.58 ± 2.13a	40.28 ± 0.63bc	49.04 ± 0.90e	
Antioxidant	6.22 ± 2.28a	43.79 ± 1.13b	54.58 ± 1.31d	6.53 ± 0.67a	44.5 ± 1.5b	49.24 ± 1.35c	5.89 ± 1.47a	43.34 ± 0.62b	56.31 ± 0.78e	5.69 ± 4.57a	48.85 ± 1.57c	58.20 ± 0.55f	
DH	0.28 ± 0.17a	21.46 ± 1.71c	28.09 ± 1.75e	0.59 ± 0.12a	22.47 ± 0.73cd	24.77 ± 1.69cd	0.44 ± 0.05a	19.76 ± 0.75b	29.60 ± 1.65e	0.26 ± 0.11a	24.04 ± 1.36cd	30.65 ± 1.82ef	
Protein	51.81 ± 2.45bc	48.98 ± 2.45b	63.16 ± 1.11de	49.3 ± 2.89b	50.72 ± 0.89cbc	59.69 ± 0.89d	49.3 ± 2.00b	44.89 ± 1.56a	64.26 ± 0.89e	48.98 ± 2.48b	55.13 ± 1.78c	69.15 ± 1.11f	
Fat	5.72 ± 1.01	1.2 ± 0.14a	1 ± 0.28a	5.88 ± 2.99	1.35 ± 0.35a	1.02 ± 0.23a	5.49 ± 0.70	1.25 ± 0.5a	0.89 ± 0.25a	5.52 ± 2.12	0.97 ± 0.47a	0.68 ± 0.13a	
Ash	7.00 ± 2.83b	4.55 ± 0.35a	4.85 ± 0.35a	6.5 ± 0.71ab	6.8 ± 0.69ab	7.04 ± 1.06b	8.00 ± 1.41bc	5.05 ± 0.64a	5.5 ± 0.7a	7.00 ± 1.25b	8.56 ± 0.78c	8 ± 0.17c	
Water	8.38 ± 0.74ab	8.39 ± 0.74ab	7.82 ± 0.55a	8.64 ± 0.98ab	8.63 ± 0.99ab	8.24 ± 1.06ab	8.41 ± 0.67ab	8.41 ± 0.67ab	7.25 ± 1.06a	9.00 ± 0.71b	9.01 ± 0.71b	7.85 ± 1.2a	
Notes:

* Control was done without pH adjustment (pH 6.4).

** Control time for hydrolysis.

Means in the same row with different superscripts (a–f) are significantly different (p < 0.05).

Yield of FPH

Yields of FPH ranged from 4.96 ± 0.72% to 49.0 ± 0.9%. The highest yield (49.0 ± 0.9%) was obtained at pH 9 after 24 h of hydrolysis The lowest yield (4.96 ± 0.72%) was obtained at pH 7 and 0 h of hydrolysis. The result suggested that pH, duration of hydrolysis, and its interaction significantly affected the yield (p < 0.05).

Antioxidant activity

The highest antioxidant activity (58.20 ± 0.55%) was obtained after 24 h hydrolysis at pH 9. FPH showed the lowest antioxidant activity (5.69 ± 4.57%) at pH 9 and 0 h of hydrolysis. Both pH and duration of hydrolysis significantly affected the antioxidant activity (p < 0.05).

Proximate composition

pH and hydrolysis time significantly affected (p < 0.05) all proximate parameters. The highest protein content (69.15 ± 1.11%) was obtained at pH 9, with 24 h of hydrolysis time. The fat content of parrotfish head FPH ranged from 0.68 ± 0.13% to 5.882.99%; the highest fat content was obtained at pH 5 with 0 h of hydrolysis time and the lowest fat content was obtained at pH 9 with 24 h of hydrolysis time.

The ash content of the FPH of parrotfish head ranged from 4.55 ± 0.35% to 8.60 ± 0.78%; the highest ash content was obtained at pH 9 with a 12 h hydrolysis time, while the lowest was observed in the control treatment (pH 6.4) with a 12 h hydrolysis time (4.60 ± 0.35%). ANOVA analysis revealed that different pH treatments resulted in significantly different results (p < 0.05). The water content of the parrotfish FPH ranged from 7.25 ± 1.06% to 9.01 ± 0.71%; the highest water content was obtained at pH 9 with a 12 h hydrolysis time (9.01 ± 0.71%), while the lowest water content was obtained at pH 7 with a 24 h hydrolysis time (7.25 ± 1.06%).

Degree of hydrolysis

The essential properties of FPH rely on the DH of the process. A high DH can be used as an indicator of effective hydrolysis. The result of DH analysis ranged from 0.26 ± 0.11% to 30.65 ± 1.82%. The highest DH was observed at pH 9 after 24 h hydrolysis, while the lowest DH was obtained at pH 7 after 2 h hydrolysis. Both pH and duration of hydrolysis significantly affected (p < 0.05) DH.

Optimum conditions for preparation of FPH

The optimum conditions for parrotfish FPH production were analyzed using the RSM, based on the yield, antioxidant activity, protein, fat, water, ash, and DH of the FPH. The overlaid contour plot as a result of RSM analysis is shown in Fig. 3.

Figure 3 Overlaid contour plot for optimum FPH.

Based on Fig. 3, it was apparent that pH 8–9 and 21.5–24 h of hydrolysis were considered the optimum conditions for producing FPH. As the longer hydrolysis time gave better FPH characteristics, pH 9 and 24 h hydrolysis were considered the optimum conditions for generating FPH with the best characteristics from the head byproduct of parrotfish.

SDS–PAGE analysis

SDS–PAGE analysis was carried out to observe the molecular weight range of the FPH obtained under optimum conditions (pH 9; 24 h hydrolysis). The result showed that the molecular weight of FPH ranged from 18.05 kDa to 75.89 kDa (Fig. 4).

Figure 4 Molecular weight distribution of parrotfish FPH.

(A) Sample (pH 9 and 24 h). (M) Molecular weight of protein standard.

Amino acid composition

The amino acid composition of the FPH from parrotfish heads extracted at pH 9 with 24 h hydrolysis was compared to the FPH from tuna heads (Bougatef et al., 2012) and commercial FPH (International Quality Ingredients (IQI), 2005) (Table 3).

Table 3 Comparison of amino acid composition of several FPH.

No.	Amino acids	FPH from parrotfish head (%)	FPH from tuna (%)	Commercial FPH (%)	
1.	L-Ser	1.81	5.18	4.90	
2.	L-Glu	14.43	11.20	14.00	
3.	L-Phe	5.53	06.18	3.70	
4.	L-Ile	4.34	4.83	4.00	
5.	L-Val	5.38	7.49	4.90	
6.	L-Ala	7.41	2.88	7.30	
7.	L-Arg	6.12	11.53	6.80	
8.	L-Gly	7.63	3.32	11.00	
9.	L-Lys	8.3	10.23	7.50	
10.	L-Asp	11.06	9.91	9.50	
11.	L-Leu	8.48	6.48	6.50	
12.	L-Pro	5.64	3.62	–	
13.	L-Tyr	4.22	5.44	2.90	
14.	L-Thr	6.80	2.17	4.40	
15.	L-His	2.85	9.52	2.60	
Total Essential Amino Acid	41.69	46.90	42.70	
Total Hydrophobic Amino Acid (HAA)	41.00	36.92	29.30	

Parrotfish FPH consists of essential amino acids (histidine, threonine, valine, isoleucine, leucine, phenylalanine, and lysine) and non-essential amino acids (aspartic acid, glutamic acid, serine, arginine, glycine, alanine, tyrosine, and proline).

Discussion

The yield of FPH from parrotfish heads is higher than that obtained from Yellow-Spotted Trivaly fish heads (71.77%) (Tawfik, 2009), tuna fish heads (9.85%) (Parvathy et al., 2018), Grouper fish head (71.20) (Tawfik, 2009). The antioxidant activity of parrotfish heads was much lower than that derived from Catla fish heads (77.92%) (Elavarasan, Kumar & Shamasundar, 2014). Heating can be applied to increase the yield, because it allows water unbound to materials to dissipate (Khan et al., 2017). Furthermore, the longer the hydrolysis time, the higher the yield. Kim (2013) stated that the FPH yield increases as a function of time of hydrolysis until its reaches a stationary phase. The highest yield obtained in this study (49.00 ± 0.9%) was lower than that of tuna (60.73%) (Ramakrishnan et al., 2013) and codfish (75%) (Himonides, Taylor & Morris, 2011).

Compared to FPH from the heads of catfish and mackerel, the antioxidant activity of FPH from parrotfish heads was still lower (Le Vo et al., 2016; Ediriweera, Aruppala & Abeyrathne, 2019). The size of peptides and the composition of free amino acids affect the antioxidant activity of FPH. The longer the hydrolysis time, the more abundant free amino acids become. Hydrophobic amino acids such as Pro, Leu, Ala, Trp, and Phe will increase antioxidant activity. In addition, Tyr, Met, His, and Lys are able to act as antioxidants (Le Vo et al., 2016).

The protein content of parrotfish head FPH was higher than that obtained from catfish heads (39.03%) (Utomo, Suryaningrum & Harianto, 2014), tuna heads (28.39%) (Ramakrishnan et al., 2013), but still lower than commercial FPH (73–75%) (International Quality Ingredients (IQI), 2005). According to Nurdiani et al. (2016), the protein content of FPH can be influenced by the amount of water dehydrated from the material. (Windsor, 2001) has categorized FPC/FPH into three types; type A (protein content is more than 80%), type B (protein content is less than 80%), and type C (low quality). Based on its protein content, FPH from parrotfish heads could be classified as a type B hydrolysate.

The fat content of parrotfish head FPH was lower than the FPH from croaker fish head waste (5.1 ± 4.0%) (Amorim et al., 2016) and commercial FPH (19–22%) (International Quality Ingredients (IQI), 2005). The low-fat content of parrotfish head FPH was due to the low-fat content in fish head raw materials (3.92%). According to Bechtel (2003), the fat content in hydrolysate products is influenced by the characteristics of the raw materials used and the process of separating fat after hydrolysis. The fat was separated mechanically during the centrifugation process.

The ash levels were higher than that of cod head waste FPH (1%) (Himonides, Taylor & Morris, 2011), but still met commercial FPH standards (4–7%) (International Quality Ingredients (IQI), 2005). The ash content in FPH tends to increase with an increasing amount of buffer (HCl and NaOH) added. According to Salamah, Nurhayati & Widadi (2011), high ash content in FPH was a result of the addition of alkali compounds, such as NaOH, or acid compounds, such as HCl, in the process of protein hydrolysis. Mixing acid and alkali compounds in the protein hydrolysate solution will cause the formation of salt compounds, which increases the ash content in protein hydrolysates. The water content of parrotfish head FPH was higher than that of cod head FPH (5%) (Himonides, Taylor & Morris, 2011) and commercial FPH standards (3–5%) (International Quality Ingredients (IQI), 2005).

The optimization result indicated that the best FPH would be produced from pH 9 and a 24 h hydrolysis time. pH 9 has previously been recorded as the best pH for hydrolyzing fish byproducts (Singh & Benjakul, 2018). One parameter that should be considered during this optimization process is the low protein content; the protein content was below commercial FPH (International Quality Ingredients (IQI), 2005). This result was also corroborated by the DH result. Norma et al. (2005) and Hau et al. (2018), reported that a longer incubation time increased the DH.

The DH of parrotfish hydrolysate was higher than that of Nile fish heads (14.3%) (Srikanya et al., 2017) and kurisi byproducts (15%) (Gajanan, Elavarasan & Shamasundar, 2016). This is possibly due to the high level of endogenous parrotfish head proteases. For the first two hours, the DH was similar from the result from Herpandi et al. (2012) and Herpandi et al. (2013), which use commercial enzymes for the hydrolysis. However, our result was lower in the 3rd h of hydrolysis. It is clear that the enzyme plays an important role in the DH. Furthermore, the physical structure and protein molecules, which exist in the sample, were affecting the DH (Kanu et al., 2009).

The DH affects protein molecular weight and amino acids. FPH from Nile fish had a wider range of molecular weight (14.4–116 kDa) (Tejpal et al., 2017) than that obtained in this study. The dominance of small peptides will increase the potency of the FPH as a bioactive substance.

The total essential amino acids of FPH from parrotfish heads (41.69%) approached the commercial standard of FPH (42.70%) (International Quality Ingredients (IQI), 2005), but was still lower than FPH from tuna heads (46.90%) (Bougatef et al., 2012). According to Chobert, Bertrand-Harb & Nicolas (1988), the content of essential amino acids indicates the potential of hydrolysates to serve as a useful source of nutrition. The difference in amino acid composition between hydrolysates depends on differences in enzyme specificity and hydrolysis conditions.

The total hydrophobic amino acid content of parrotfish FPH (41%) was higher than the FPH from tuna heads (36.92%) (Bougatef et al., 2012) and commercial FPH (29.30%) (International Quality Ingredients (IQI), 2005). The amino acid composition can also affect the functional properties of FPH, such as the nature of the antioxidant activity. According to Zainol et al. (2003), hydrophobic amino acids (alanine, leucine, and proline) have been shown to have free radical quenching activities. Hydrophobic aromatic amino acids (tyrosine and phenylalanine) can also function as antioxidants by donating electrons.

Analysis of amino acids can determine the quality of FPH manufactured, specifically from the ratio of amino acids contained in these proteins (Nurilmala, Nurhayati & Roskananda, 2018). According to Annisa, Darmanto & Amalia, 2012, the amino acids in each fish species vary depending on internal and external factors. Internal factors include fish species, sex, age, and the reproduction phase of the fish, while external factors are typically environmental.

Conclusions

Characteristics of FPH from the heads of parrotfish (C. sordidus) were affected by the ratio of minced fish head: dH2O, pH, and duration of hydrolysis. The yield, antioxidant activity, protein content, ash content, and DH of the FPH were dependent on pH and time of hydrolysis. The optimum conditions for the production of FPH from parrotfish heads include a minced head: dH2O ratio of 1:2 (w/v), at pH 9, with a 24 h hydrolysis time. The process generated an essential amino acid profile of 41.69%. To the best of our knowledge, this is the first report on the added value of C. sordidus heads.

Supplemental Information

Supplemental Information 1 Raw Data.

Click here for additional data file.

Additional Information and Declarations

Competing Interests

Author Contributions

Data Availability

The authors declare that they have no competing interests.

Asep A. Prihanto conceived and designed the experiments, performed the experiments, analyzed the data, contributed reagents/materials/analysis tools, prepared figures and/or tables, authored or reviewed drafts of the paper, approved the final draft.

Rahmi Nurdiani conceived and designed the experiments, performed the experiments, analyzed the data, authored or reviewed drafts of the paper, approved the final draft.

Annas D. Bagus performed the experiments, analyzed the data, contributed reagents/materials/analysis tools, prepared figures and/or tables, approved the final draft.

The following information was supplied regarding data availability:

The raw data is available in the Supplemental File and at figshare: Prihanto, Asep Awaludin; Nurdiani, Rahmi; Bagus, Annas Dwi (2019): Raw Data: Production and characteristics of fish protein hydrolysate from parrotfish (Chlorurus sordidus) head. figshare. DOI 10.6084/m9.figshare.10264805.v2.

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
