# Peer review of "Production and characteristics of fish protein hydrolysate from parrotfish (Chlorurus sordidus) head"

_PeerJ, doi:10.7717/peerj.8297_

## Round 0.1 · original submission · Major Revisions

Please read carefully and present a revised version. In particular, the description of a single sample (n=1) is worrisome. Legends for tables and figures require more detail, experimental methods as well.

·

Basic reporting

Very good English. Sufficient references. Data supported with good discussion. Good tables and figures presentation.

Experimental design

The research question stated clearly. Information on sample preparation and analysis needs to improve.

Validity of the findings

The finding useful to use a head by-product from parrotfish.

Additional comments

Abstract:
Line 24: This research is less correlation with endogenous enzymes, more o the effect of pH, time and sample ratio on characteristics of FPH. Suggested to change the term "endogenous enzymes" to "different pH, time and sample ratio"
Line 32. Since the level of essential amino acids of the sample is lower than the provided comparison (FPH from tuna and Commercial FPH), the statement "with a high level of essential amino acids" needs to delete except authors can come with some evidence.

Introduction:
Line 42: FAO, 2015. Please change with updated data (2017 or 2018) - if available. Any data on parrotfish production and utilization in Indonesia?
Line 45: 19% for general fish head or dedicated only for parrotfish head?
Line 46. Please also highlight the utilization of head scale and bone fish-by products as material for gelatin production - (Fish bone and scale as a potential source of halal gelatin. Journal of Fisheries and Aquatic Science, 2011, 6(4), pp. 379-389).
Line 59: Please also refer to additional reference for the production of FPH using fisheries by-products - Protein quality of hydrolyzed dark muscle protein of skipjack tuna (Katsuwonus pelamis) Turkish Journal of Fisheries and Aquatic Sciences, 2016, 16(1), pp. 177-186.


Materials & Methods
Line 70. Need to state the size of the parrotfish head sample (weight or dimension).
Line 75: Need to state the procedure to mince the head of parrotfish.
Line 77: Need to state the exact temperature of the experiment room.
Line 137. Probably amino acid analysis uses the Acid Hydrolysis method for sample hydrolysis preparation since data of amino acid tryptophan, methionine and cystine are not presented.

Discussion:
Line 246. Need to provide discussion on the proximate composition of Parrotfish head. Comparison with the head of Nile fish not suitable since not belong to fresh head, but consider as dried head as indicated by lower water (moisture) content.
Line 261. Please check why the total component of protein, fat, ash and water on (dried) FPH at Table 2 is not near to 100%, just around 36 - 55%. What components contributes to the resat of 45 - 64%?
Line 285. Please check and compare the resulted degree of hydrolysis with the references - Degree of hydrolysis and free tryptophan content of Skipjack Tuna (Katsuwonus pelamis) protein hydrolysates produced with different type of industrial proteases. 2012. International Food Research Journal
19(3), pp. 863-867 and Optimizing the enzymatic hydrolysis of skipjack tuna (Katsuwonus pelamis) dark flesh using Alcalase® enzyme: A response surface approach. 2013. Journal of Fisheries and Aquatic Science, 8(4), pp. 494-505.

Conclusion.
Good and Sufficient.

References:
Please refer to the above-provided comments for additional references.

Table 3:
Need to mention that the Total Essential Amino Acid refer to Total Essential Amino Acid without tryptophan and methionine.

Reviewer 2 ·

Basic reporting

The work entitled “Production and characteristics of fish protein hydrolysate from parrotfish (Chlorurus sordidus) head” proposes to determine the characteristics of protein hydrolysates extracted at different pHs and hydrolysis duration periods. The information is not original, though if the species has important values of wastes, would be important to study, however, the authors don't discuss this topic.

Experimental design

The experimental design is incorrect because the authors don't include a control treatment, as "initial time" that corresponds to raw material without hydrolysis. It is important to have a control treatment to compare the results; sometimes, the raw material has high antioxidant activity for example and peptides of different molecular weights. In this work, there is no evidence of control treatment for then some results can’t be discussed.

Validity of the findings

Some results are too confusing. Figure 4, for example, has not well defined and is very incomplete. I think that is essential to show the electrophoresis for different hydrolysates and the raw protein without hydrolysis to compare, as control. Furthermore, I don't sure that proteins with molecular weight from 18.05 kDa to 75.89 kDa were the products after hydrolysis of 24 hs.

Additional comments

The manuscript is inconsistent. The principal error is the lack of initial control in the experimental design. Furthermore, the discussion section needs some argue about the importance of to use wastes of fisheries.

·

Basic reporting

1- Topic of this research is important, since many different groups around the world are working on it, seeking how to use fish by-products, waste and efluents from the fishing industry as sources of bioactive compunds with potential value added. However, the writting of the manuscript is confuse and difficult to follow.
2- List of references is ok as the number and year of publicación. However, introduction is poor because discusión about previous research is short, lite and unsufficient.
3-The main goal and hypothesis of research were not mentioned.

Experimental design

1- Material and methods are poorly described and needs better work, especielly in the statistical part. Experimental statistical design do not mention análisis of variance for Figure 2 and Table 2, however this statistical tool do not apply, because the experiment design only has one repetition (n=1). Analyses were carried out in triplicate, but n=1. Overall, Figures and Tables are poorly described.
2- Nothing was said about endogenous enzymes and how experimental pHs and time of hydrolysis were chosen.
3- Información about initial conditions of fish heads were not mentioned.

Validity of the findings

1- Difficult to have conclusive results, since the experiment has only one sample (There were no replicates. Besides, each time an hydrolisis is carried out, results could be different.
2- The Authors only describe results, and did not make any in deep análisis. For instance, how many polypeptides are in each hydrolizate? . Aminoacid profile is not useful. Primary estructure could be better. Conclusions are poor, if any.

Additional comments

Manuscript must be rewritten, with in deep análisis of data, and justify why only one sample was taken. Material and methods must be better described, mainly the statistical part.

---

## Round 0.2 · accepted · Accept

The authors have addressed the concerns of the reviewers. The paper has improved and the conclusions, are supported by the data.